# Healthcare professionals' perception and COVID-19 vaccination attitudes in North-Western Ghana: A multi-center analysis

**Augustine Ngmenemandel Balegha**[1]*, **Suburu Abdul-Aziz**[2◉], **Louis Mornah**[3◉]

**1** Department of Paediatrics, St. Theresa's Hospital, Nandom, Upper West Region, Ghana, **2** Department of Internal Medicine, Wa Municipal Hospital, Wa, Upper West Region, Ghana, **3** Department of Nutrition, Wa Municipal Hospital, Wa, Upper West Region, Ghana

◉ These authors contributed equally to this work.
* bnaugustine@gmail.com

## Abstract

### Introduction

The novel coronavirus SARS-CoV-2 causes Coronavirus Disease 2019 (COVID-19). Vaccination has been identified as one of the most effective strategies for combating COVID-19. Positive perceptions and attitudes of HCPs towards the COVID-19 vaccination are essential to vaccine uptake and adherence. However, the perceptions and attitudes of HCPs towards the COVID-19 vaccination remain largely unexplored. We therefore assessed healthcare professionals' perceptions, attitudes, and predictors of their attitudes towards COVID-19 vaccination in the Wa Municipality, Upper West Region of Ghana.

### Methods

In 2023, from January 16th to February 28th, we administered a multi-centre e-survey to a cross-section of 403 healthcare professionals in Wa Municipality of the Upper West Region, Ghana. We used STATA version 13 to analyze the data. Frequencies, percentages, and composite scores were used to assess perceptions and attitudes towards the COVID-19 vaccination. Hierarchical binary logistic regression modeling was then used to determine the predictors of attitudes towards the COVID-19 vaccination.

### Results

The healthcare professionals had positive perceptions [6.00; IQR = 4.00–7.00] and attitudes [5.00; IQR = 4.00–5.00] towards theCOVID-19 vaccination. Positive perception [aOR = 1.81; 95% CI = 1.14–2.87, $p < 0.05$], female sex [aOR = 0.58; 95% CI = 0.35–0.97, $p < 0.05$], marital status [aOR = 1.94; 95% CI = 1.20–3.12; $p < 0.01$], having a bachelor's degree or higher [aOR = 2.03; 95% CI = 1.01–4.12; $p < 0.05$], and working in the Wa North sub-Municipal area [aOR = 0.22; 95% CI = 0.05–0.96; $p < 0.05$] were statistically significantly associated with attitudes towards COVID-19 vaccination.

**Funding:** The author(s) received no specific funding for this work.

**Competing interests:** The authors have declared that no competing interests exist.

## Conclusion

The healthcare professionals' perceptions and attitudes towards the COVID-19 vaccination were positive but suboptimal. We recommend regular education on COVID-19 vaccine benefits, safety, and efficacy. Enabling the work environment and addressing vaccine availability and accessibility for healthcare professionals should also be prioritized. These measures should particularly focus on female, single healthcare professionals who possess below a bachelor's degree and are working in the Wa North sub-municipal area.

## Introduction

The novel coronavirus SARS-CoV-2 causes Coronavirus Disease 2019 (COVID-19). Since its emergence in late 2019, the disease has caused significant health, social, and economic impacts worldwide [1]. Globally, the number of confirmed cases of COVID-19 is well over 700 million, with over 6 million deaths. Africa accounts for about 1.3% of the confirmed cases of COVID-19. To combat the spread of the virus, vaccination has been identified as one of the most effective measures [2].

The COVID-19 vaccine has therefore been developed and authorized for use worldwide [2]. Globally, over 13 billion doses of the COVID-19 vaccines have been administered [2]. As of June 30, 2023, Ghana had administered almost 26 million doses of the COVID-19 vaccine [3]. Despite the global efforts to achieve wider COVID-19 vaccination coverage to increase herd immunity, many persons (including healthcare professionals [HCPs]) remain unvaccinated, partly due to their perceptions and attitudes towards the COVID-19 vaccine [4,5].

HCPs include a wide range of cadres, including doctors, nurses, pharmacists, medical laboratory scientists, other allied health professionals, and administrative staff who provide healthcare services to patients or clients. In this study, the perceptions of HCPs on COVID-19 vaccination encompass their beliefs, opinions, impressions, and feelings about the importance, efficacy, and safety of the COVID-19 vaccine. Attitudes towards the COVID-19 vaccination in this study refer to the HCPs' intentions and predispositions towards the COVID-19 vaccination. As frontline workers, HCPs play a critical role in controlling the spread of the virus and administering the vaccine [5]. Their attitudes and perceptions towards the COVID-19 vaccine are therefore vital in promoting vaccine uptake and adherence [5,6]. Several studies have examined HCPs' attitudes and perceptions towards the COVID-19 vaccination globally.

In a Bangladeshi cross-sectional community survey, Islam et al. [7] reported positive attitudes towards the COVID-19 vaccination. Similarly, a public survey in Jordan reported positive attitudes towards the COVID-19 vaccination [8]. Likewise, studies conducted in sub-Saharan Africa portray mixed findings. In Ethiopia, Adane et al. [9] reported that the HCPs had a good perception (60.5%) and positive attitudes (52.3%) towards the COVID-19 vaccination. In Nigeria, Abduljaleel [10] reported a poor perception (24.3%) of COVID-19 vaccination among nurses. In Ghana, there is a paucity of studies examining the perceptions and attitudes of HCPs towards COVID-19 vaccinations. These few studies have largely concentrated on COVID-19 vaccine acceptance among HCPs and not their overall attitudes towards COVID-19 vaccination. Additionally, no study linking perceptions, individual-level factors, and contextual factors to the attitudes of HCPs has been conducted. Moreover, studies on COVID-19 in Ghana have mostly focused on southern Ghanaian cities, to the neglect of relatively resource-poor regions like the Upper West Region (UWR) of Ghana. Therefore, a

knowledge gap exists in terms of healthcare professionals' perceptions and attitudes towards COVID-19 vaccination in the Wa Municipality, Upper West Region of Ghana.

Therefore, this study assessed the perceptions, attitudes, and predictors of attitudes towards COVID-19 vaccination among HCPs in the Wa Municipality of the UWR of Ghana. Such research is crucial in designing effective vaccination campaigns and addressing concerns or barriers to HCPs' perceptions and attitudes towards COVID-19 vaccination.

## Methods

### Conceptual framework of the study

We present the conceptual framework of the study in **Fig 1**. Based on empirical evidence, we propose a model dubbed the Perception-Individual-Context-Attitude-Acceptance (PICA-A) framework. In this framework, we theorize that perception, individual level, and contextual level factors independently predict attitudes, which culminate in acceptance of an essential service. As shown in Fig 1, we hypothesize that the health professionals' perceptions (operationalized as poor perception and good perception), individual level (sex, age, marital status,

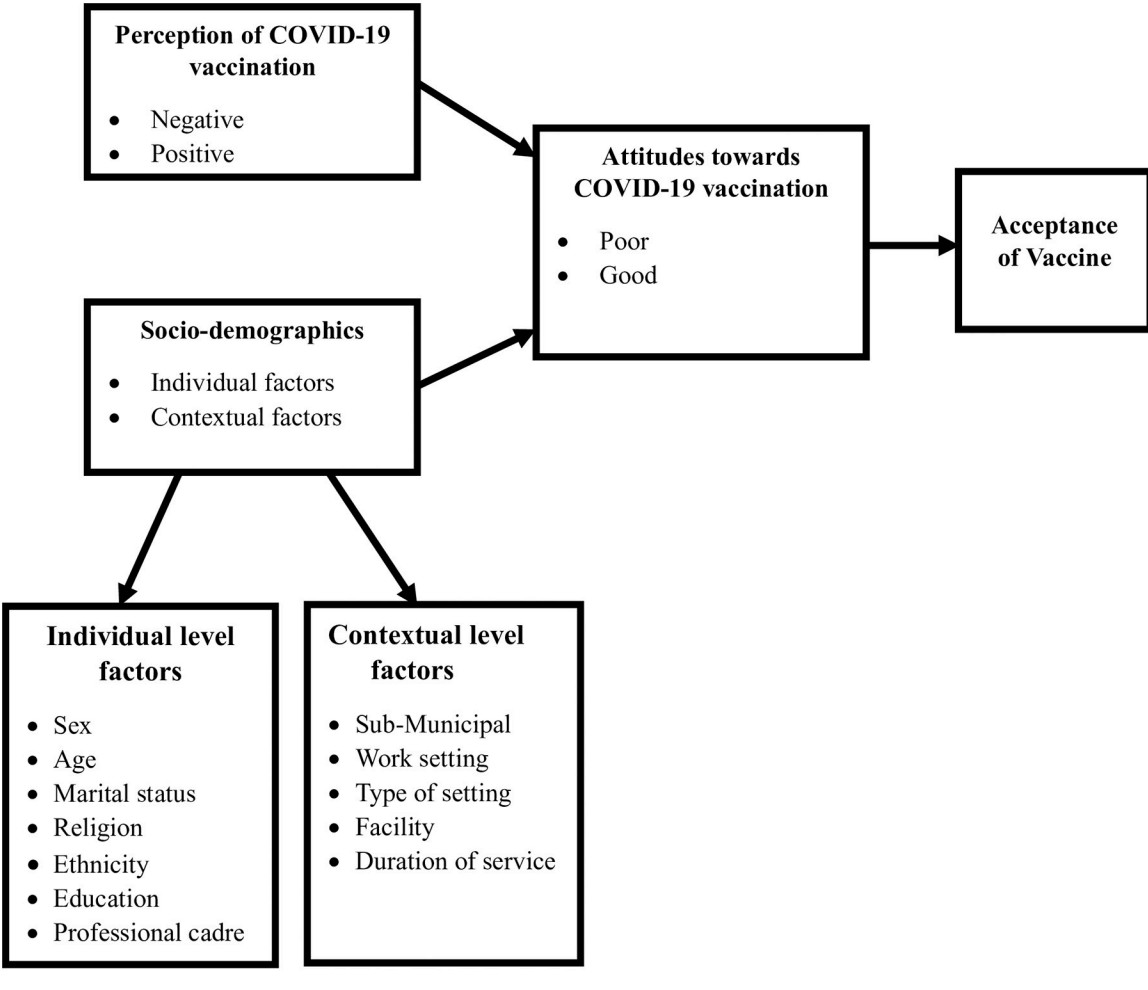

*PICA-A model*

**Fig 1. Conceptual framework of the study.**

religion, ethnicity, and education), and contextual level (professional cadre, sub-municipal, work setting, type of setting, facility, and duration of service) factors influence their attitudes towards COVID-19 vaccination (operationalized as poor and good attitudes towards COVID-19 vaccination). The outcome variable in this study is the attitudes of health professionals towards the COVID-19 vaccination. Perceptions of COVID-19 vaccination are the key explanatory variable, while individual and contextual factors are the other explanatory factors. However, although we postulate a direct link between attitudes and acceptance of the COVID-19 vaccination among health professionals, such a phenomenon is beyond the scope of this study.

## Study approach, design and setting

The study employed a quantitative approach and a multicenter cross-sectional design. This offered the opportunity to sample and analyze data on perceptions and attitudes towards COVID-19 vaccination among a cross-section of health professionals in the Wa Municipality. Wa Municipality is one of the 11 districts of the UWR of Ghana. The municipality shares administrative boundaries with Nadowli District to the north, Wa East District to the east and west, and Wa-West District to the south. It lies within latitudes 1˚40'N to 2˚45'N and longitudes 9˚32'W to 10˚20'W. Wa Municipality has its capital in Wa, which also serves as the regional capital of the UWR. It has a land area of approximately 579.86 square kilometers, which accounts for about 6.4% of the total land mass of the region. The municipality has a population of 200,672, representing 22.3% of the regional population [11]. In terms of healthcare delivery, the municipality, which is predominantly urban, is divided into ten (10) sub-municipals, namely, Bamahu, Busa, Charia, Charingu, Dobile, Kambali, Kperisi, Kpongu, Wa North, and Wa South. The municipality has a regional hospital, a municipal hospital, 4 private hospitals, 10 health centers, 30 functional CHPS zones, 132 communities, and outreach points with 264 community-based agents who support the sub-municipal staff to carry out community-based activities [12]. These health facilities deliver both curative and preventive health services. As of May 21, 2023, UWR had recorded 996 cases of the 171,740 confirmed cases of COVID-19 in Ghana [3]. However, statistics of COVID-19 infection rates in Wa Municipality, as well as its sub-municipal breakdown, are not readily accessible.

## Study population

The study population included all the HCPs in the Wa municipality, who provide diverse healthcare services within the municipality. HCPs were studied because they constitute the frontline workers in the fight against COVID-19 and are significantly exposed to COVID-19 as an occupational hazard. In addition, healthcare professionals' perceptions and attitudes towards the COVID-19 vaccination have the potential to influence the general population's acceptance of the vaccine. In this study, HCPs were grouped into six categories: physicians, nurses, pharmacy technicians, medical laboratory staff, other allied health staff (paramedics, radiology technicians, and ancillary staff), and administrative staff. The inclusion criteria for the selection of participating HCPs were that the HCPs had been working in any of the health facilities within the Wa municipality for at least six months prior to the study. However, HCPs on study leave, sick leave, maternity leave, or annual leave were excluded from the study. These categories of HCPs were excluded because they were no longer in direct contact with COVID-19 activities and may not be privy to information regarding COVID-19 operations within the service as compared to those who were at post.

## Sampling

We used Cochran's formula to estimate the sample size of the study. According to Cochran [13], the sample size of a population may be estimated using: $n = \frac{z^2 pq}{d^2}$; where n is the sample

size, z = 1.96 at 95% confidence level, p = the proportion of healthcare professionals with good attitudes towards the COVID-19 vaccination = 0.5 (largest assumed estimate for p), q = 1-p, and d = estimated margin of error = 0.05.

Therefore, n $= \frac{(1.96)^2 \ x \ 0.5 \ x \ (1-0.5)}{(0.05)^2} = 384$

Applying a 10% non-response rate to account for incomplete questionnaires and non-responses and ensuring that an absolute number of the different cadres of HCPs were sampled, the sample size was approximated to be 420. A multistage cluster sampling technique was used to select the study participants. First, we stratified the municipality based on the existing ten administrative sub-municipal clusters of healthcare delivery systems. Each cluster was then stratified into six healthcare professional cadres (administrative staff, pharmacists, biomedical scientists, other allied health professionals, and nurses' physicians). Seven HCPs were then drawn at random from each cadre of HCPs using balloting. Where a particular sub-municipal lacked the required number of a particular professional cadre to be sampled, other available professional cadres were sampled in proportion to the sample size of the available cadres. This strategy was applied to ensure that a representative sample from each cluster was recruited for the survey.

## Ethical consideration

We obtained ethical clearance *(GHS-ERC: 037/08/22)* from the Ghana Health Service Ethics Review Committee. Formal approval was also obtained from the regional director and the Wa Municipal Director of Health Services. Research protocols and objectives were explained to study participants prior to the study. Written, informed consent was obtained from each participant. Study participants were assured of data confidentiality and anonymity. Study participants were reserved the absolute right to withdraw at any point in the research. The study was conducted in accordance with the Declaration of Helsinki.

## Data collection techniques and tools

We collected primary data using a well-structured questionnaire (**S1 Questionnaire**) adapted from several relevant pieces of literature [9,14,15]. The questionnaire encompassed three sections. Section A assessed 12 socio-demographic characteristics of the respondents. These socio-demographic characteristics comprised 7 individual-level factors (sex, age, marital status, religion, ethnicity, educational background, and professional cadre) and 5 contextual-level factors (sub-municipality, work setting, facility type, location of facility, and duration of service) for the health professionals. Section B measured the perception of health professionals on COVID-19 vaccination using 10 questions on knowledge of an infected person, perception of contact with an infected person, fear of infection, reality of COVID-19, protective measures at work, safety of vaccine types, COVID-19 vaccine preventing spread among patients, COVID-19 vaccine preventing spread among hospital workers, safety and effectiveness of vaccines, and immune-potency of COVID-19 vaccines. Section C assessed their attitudes towards the COVID-19 vaccination based on six questions covering prior COVID-19 infection, prior vaccination with the COVID-19 vaccine, alternative COVID-19 preventive measures, prioritizing at-risk groups, prioritizing health workers, and following governmental vaccine guidelines. The questionnaire was created on an online platform (KoboCollect) in English, from which a re-directory link to the questionnaire was shared with participants who were sampled from the target population. Data were collected from the participants from January 16th, 2023, to February 28th, 2023.

## Validity and reliability of the instrument

Based on the expertise of three consultants, the face validity and content validity (content validity index, CVI = 0.78) of the data collection instrument were examined to ensure that the questions were appropriate for the local context and in line with the objectives of the study. We then pretested the instrument among 30 HCPs working in healthcare facilities in the Jirapa Municipality. We used the feedback from the pretest to restructure the questions to ensure clarity and purpose. We then determined the construct validity of the instrument using confirmatory factor analysis. Our confirmatory factor analysis results depicted good model fitness indices of $\chi 2$ (34) = [53.800, $p$ = 0.017]; root mean squared error of approximation (RMSEA) = 0.038; comparative fit index (CFI) = 0.937; Tucker-Lewis index (TLI) = 0.916; and standardized root mean squared residual (SRMR) = 0.039 [16]. We also tested the reliability of the instrument using Cronbach's alpha ($\alpha$). The computed values were acceptable. The overall $\alpha$ was 0.64 for all scales of the instrument, while the $\alpha$ for the individual scales of socio-demographic factors, perception, and attitudes were $\alpha_s$ = 0.67, $\alpha_p$ = 0.64, and $\alpha_a$ = 0.60, respectively.

## Study variables

The outcome variable in this study was the attitude of healthcare professionals towards the COVID-19 vaccination. The main explanatory variable is the perception of the HCPs on COVID-19. Other explanatory variables included in this study are the socio-demographic characteristics of the respondents, which were classified into individual-level factors and contextual-level factors. Table 1 presents the definition, measurement, and reference categories of the outcome and explanatory variables.

## Data processing and analysis

Data (S1 Data) collected in the field were double-cross-checked, cleaned of incomplete forms, coded, and analyzed using the STATA computer program, version 13. The analysis of the data was done in four steps.

In the first step, the socio-demographic characteristics (both individual and contextual factors), perceptions, and attitudes of the respondents towards the COVID-19 vaccination were analyzed using frequencies and percentages. In the second step, we assigned a score of 1 to correctly answered questions and no score for wrongly answered questions for the perceptions and attitudes sub-scales. We then summed the scores to compute composite scores for perceptions and attitudes towards the COVID-19 vaccination. The minimum and maximum attainable scores for perception were 0 and 10, while those for attitudes were 0 and 6, respectively. We then computed summary statistics (minimum, maximum, range, mean with standard deviation, median with interquartile range, skewness, and kurtosis) for both sub-scales. Perception and attitudes towards the COVID-19 vaccination were then re-categorized into negative and positive perceptions and poor and good attitudes, respectively. Since data for both perceptions and attitudes towards COVID-19 vaccinations were negatively skewed, the median was used as the measure of central tendency for re-categorization. In the third step, multi-collinearity was tested among the socio-demographic factors using the variance inflation factor (VIF). There was no multi-collinearity among the socio-demographic factors, as VIF was < 10 (mean VIF = 1.28; minimum VIF = 1.03; maximum VIF = 1.66). In the fourth step, individual factors (perceptions, individual, and contextual factors) associated with attitudes towards COVID-19 vaccinations were assessed using chi square tests at the bivariate level. Significant factors at the bivariate level of analysis were then included in a multivariate logistic regression model to eliminate confounding and spurious associations. Binary logistic regression analysis was performed because the outcome variable of the study (attitudes towards

**Table 1. Description of the variables of the study.**

| Variables | Definition | Measurement | Reference |
|---|---|---|---|
| **Outcome variable** | | | |
| Attitudes towards COVID-19 vaccination | Beliefs, tendencies | 0. Poor<br>1. Good | Poor |
| **Key explanatory variable** | | | |
| Perception on COVID-19 vaccination | Opinions, viewpoints | 0. Negative<br>1. Positive | Negative |
| **Socio-demographics** | | | |
| **Individual factors** | | | |
| Sex | Male or female sex assigned at birth | 0. Male<br>1. Female | Male |
| Age groups | Age at last birth day grouped | 1. 21–29<br>2. 30–39<br>3. 40–49<br>4. 50+ | 21–29 |
| Marital status | Status of being married or unmarried | 0. Unmarried<br>1. Married | Unmarried |
| Religion | Religious affiliation | 0. Islam<br>1. Christian | |
| Ethnic background | Ethnicity of respondent | 1. Dagaaba<br>2. Waala<br>3. Sissala<br>4. Akan<br>5. Others | Dagaaba |
| Educational qualifications | Highest educational attainment | 1. Certificate<br>2. Diploma<br>3. Degree and above | Certificate |
| Professional cadre | Cadre of health professional | 1. Biomedical Scientist<br>2. Pharmacist/ technician<br>3. Administrative staff<br>4. Allied health<br>5. Nurse<br>6. Physician | Biomedical Scientist |
| **Contextual factors** | | | |
| Sub-Municipal | Sub-Municipal of the respondent | 1. Bamahu<br>2. Busa<br>3. Charia<br>4. Charingu<br>5. Dobile<br>6. Kambali<br>7. Kperisi<br>8. Kpongu<br>9. Wa North<br>10. Wa South | Bamahu |
| Work setting | Government versus Private facility | 1. Private<br>2. Government | Private |
| Facility type | Level of healthcare delivery | 1. CHPS<br>2. Clinic<br>3. Health centre<br>4. Polyclinic<br>5. Municipal Hospital<br>6. Regional Hospital | CHPS |
| Current facility | Rural versus Urban | 1. Rural<br>2. Urban | Rural |
| Duration on service | Number of years in service | 0. $\leq$ 5 years<br>1. > 5 years | $\leq$ 5 years |

COVID-19 vaccination) is binary categorical with two categories (poor attitude versus good attitude towards COVID-19 vaccination).

Four binary logistic regression models were fitted. The first model was a null model, which contained none of the explanatory variables but the random intercept. Model 1 assessed the association between the outcome variable, the key explanatory variable, and individual-level variables. Model 2 involved the outcome variable, the key explanatory variable, and the contextual-level variables. The final model, model 3, contained the outcome variable, key explanatory variable, individual level, and contextual level variables. The fixed effects inferential analyses (both bivariate and multivariate) were considered statistically significant at $\rho < 0.05$ and 95% confidence intervals (CI). Model comparisons were done using pseudo-log-likelihood, pseudo-$R^2$, and Akaike's information criterion (AIC). The results of our analysis were reported based on the guidelines for Strengthening the Reporting of Observational Studies in Epidemiology (STROBE) and the Checklist for Reporting Results of Internet E-surveys (CHERRIES).

## Results

### Sociodemographic characteristics of the respondents

Table 2 presents the socio-demographic characteristics of the respondents. Of the 420 health professionals sampled, 403 of them responded to the survey, giving a response rate of 95.95%. As shown in Table 2, the majority (69.2%) of the respondents were female. The median age (IQR) of the respondents was 33 (30–39) years, with the majority (53.3%) in the 30–39 age bracket. The majority of the health professionals were married (72.7%), Christians (54.3%), and Dagaabas (38.0%). The most prevalent educational qualification among the respondents was a diploma (49.1%), and the least prevalent was a bachelor's degree or above (18.6%). The majority of the respondents were nurses (79.7%) who worked in the Wa South sub-municipal area (30.8%), within government facilities (91.6%), and in health centers (41.9%). The majority of the respondents had worked in urban facilities (88.8%) for more than 5 years (52.9%), with a mean duration of work of 7.17 (±5.36) years.

### Perception and attitudes of the healthcare professionals towards COVID-19 vaccination

Table 3 presents the HCPs' perceptions of the COVID-19 vaccination. As depicted in Table 3, less than one-third (24.6%) of the study participants knew a person infected with COVID-19 and had had prior contact (23.6%) with an infected person. The majority (68.5%) of the respondents indicated fear of the COVID-19 infection, as almost all (98.5%) of them reported that COVID-19 is real. Almost all (98.8%) of the health professionals indicated the availability of protective measures at their workplaces. As the majority (59.1%) of the respondents perceived all COVID-19 vaccines to be safe, about a third (30.8%) of them perceived Pfizer/BioNtech Comirnaty to be the safest, and Gamaleya Sputnik V the least (3.7%) in terms of safety. The majority of the respondents perceived that the COVID-19 vaccination would prevent the spread of infection among patients (73.2%) and among hospital workers (72.0%). The majority of the respondents perceive the COVID-19 vaccines to be both safe and effective (70.7%) and immunocompetent (59.1%).

Table 4 displays the results of the attitudes of the HCPs towards the COVID-19 vaccination. As shown in Table 4, 3.7% of the health professionals had ever contracted the COVID-19 infection in line with their duties. The majority of the health professionals (92.1%) had been vaccinated with Oxford AstraZeneca vaccines (38.2%) as their first line of protection. Of those vaccinated, 79.9% had completed their vaccination, thus either the Pfizer/BioNtech Comirnaty

**Table 2. Socio-demographic characteristics and healthcare professionals' attitudes towards COVID-19 vaccination (n = 403).**

| Variables | Frequency | Percentage (%) | Attitudes towards COVID-19 vaccination | | χ2 (df), p-value |
| --- | --- | --- | --- | --- | --- |
| | | | Negative (%) | Positive (%) | |
| **Sex** | | | | | **12.1 (1), _p_ = 0.001** |
| Male | 124 | 30.8 | 37 (29.8) | 87 (70.2) | |
| Female | 279 | 69.2 | 135 (48.4) | 144 (51.6) | |
| **Age** | | | | | **4.8 (3), _p_ = 0.190** |
| 21–29 | 98 | 24.3 | 50 (51.0) | 48 (49.0) | |
| 30–39 | 215 | 53.3 | 86 (40.0) | 129 (60.0) | |
| 40–49 | 79 | 19.6 | 30 (38.0) | 49 (62.0) | |
| 50+ | 11 | 2.7 | 6 (54.5) | 5 (45.5) | |
| **Marital status** | | | | | **10.1 (1), _p_ = 0.001** |
| Single | 110 | 27.3 | 61 (55.5) | 49 (44.5) | |
| Married | 293 | 72.7 | 111 (37.9) | 182 (62.1) | |
| **Religion** | | | | | **0.8 (1), _p_ = 0.360** |
| Islam | 184 | 45.7 | 74 (40.2) | 110 (59.8) | |
| Christian | 219 | 54.3 | 98 (44.7) | 121 (55.3) | |
| **Ethnicity** | | | | | **1.1 (4), _p_ = 0.897** |
| Dagaaba | 153 | 38.0 | 64 (41.8) | 89 (58.2) | |
| Waala | 123 | 30.5 | 52 (42.3) | 71 (57.7) | |
| Sissala | 29 | 7.2 | 11 (37.9) | 18 (62.1) | |
| Akan | 85 | 21.1 | 40 (47.1) | 45 (52.9) | |
| Others* | 13 | 3.2 | 5 (38.5) | 8 (61.5) | |
| **Educational Qualification** | | | | | **14.62 (2), _p_ = 0.001** |
| Certificate | 128 | 31.8 | 62 (48.4) | 66 (51.6) | |
| Diploma | 198 | 49.1 | 92 (46.5) | 106 (53.5) | |
| Degree or higher | 77 | 19.1 | 18 (23.4) | 59 (76.6) | |
| **Professional category** | | | | | **8.1 (5), _p_ = 0.151** |
| Biomedical Scientist | 18 | 4.5 | 9 (50.0) | 9 (50.0) | |
| Pharmacist/ technician | 10 | 2.5 | 4 (40.0) | 6 (60.0) | |
| Administrative staff | 5 | 1.2 | 2 (40.0) | 3 (60.0) | |
| Other allied health** | 6 | 1.5 | 2 (33.3) | 4 (66.7) | |
| Nurse | 321 | 79.7 | 145 (45.2) | 176 (54.8) | |
| Physician | 43 | 10.7 | 10 (23.3) | 33 (76.7) | |
| **Sub-Municipal** | | | | | **20.83 (9), _p_ = 0.013** |
| Bamahu | 15 | 3.7 | 3 (20.0) | 12 (80.0) | |
| Busa | 7 | 1.7 | 2 (28.6) | 5 (71.4) | |
| Charia | 10 | 2.5 | 3 (30.0) | 7 (70.0) | |
| Charingu | 5 | 1.2 | 1 (20.0) | 4 (80.0) | |
| Dobile | 115 | 28.5 | 46 (40.0) | 69 (60.0) | |
| Kambali | 50 | 12.4 | 19 (38.0) | 31 (62.0) | |
| Kperisi | 5 | 1.2 | 3 (60.0) | 2 (40.0) | |
| Kpongu | 11 | 2.7 | 2 (18.2) | 9 (81.8) | |
| Wa North | 61 | 15.1 | 39 (64.0) | 22 (36.0) | |
| Wa South | 124 | 30.8 | 54 (43.5) | 70 (56.5) | |
| **Type of work setting** | | | | | **0.3 (1), _p_ = 0.590** |
| Private | 34 | 8.4 | 17 (50.0) | 17 (50.0) | |
| Government | 369 | 91.6 | 170 (46.1) | 199 (53.9) | |
| **Type of facility** | | | | | **8.2 (5), _p_ = 0.148** |

_(Continued)_

**Table 2.** (Continued)

| Variables | Frequency | Percentage (%) | Attitudes towards COVID-19 vaccination | | χ2 (df), p-value |
|---|---|---|---|---|---|
| | | | Negative (%) | Positive (%) | |
| CHPS | 79 | 19.6 | 39 (49.4) | 40 (50.6) | |
| Clinic | 32 | 7.9 | 20 (62.5) | 12 (37.5) | |
| Health Center | 169 | 41.9 | 76 (45.0) | 93 (55.0) | |
| Polyclinic | 2 | 0.5 | 1 (50.0) | 1 (50.0) | |
| Municipal Hospital | 14 | 3.5 | 8 (57.1) | 6 (42.9) | |
| Regional Hospital | 107 | 26.6 | 43 (40.2) | 64 (59.8) | |
| **Current facility location** | | | | | **5.3 (1), $p = 0.021$** |
| Rural | 45 | 11.2 | 12 (26.7) | 33 (73.3) | |
| Urban | 358 | 88.8 | 175 (48.9) | 183 (51.1) | |
| **Duration in service** | | | | | **1.0 (1), $p = 0.322$** |
| < = 5 years | 190 | 47.1 | 95 (50.0) | 95 (50.0) | |
| >5 years | 213 | 52.9 | 92 (43.2) | 121 (56.8) | |

χ2 = Chi-square; df = degrees of freedom

* Ga, Ewe, Kasena, Moosi, Gonja

** Nutrition Officer, Disease control officers, Health information/biostatisticians

single-shot dose or the double dose of the Oxford Astra-Zeneca vaccines. The major (11.6%) barrier to completing the vaccination was the side effects of the initial shot. Reasons for not taking the vaccine included unknown long-term effects of the vaccines (6.5%), perception of vaccine-induced infertility (2.5%), and severe allergic reactions (0.2%). Meanwhile, 5.5% of the respondents reported having plans to receive the vaccine soon. The health professionals' most common alternative COVID-19 infection preventive measures were the observation of the COVID-19 protocols (89.1%), physical activity (6.0%), praying (4.5%), and traditional medicine (0.5%). The majority of the HCPs are of the view that people with chronic and severe diseases (76.9%) and health professionals (85.4%) should get priority for the COVID-19 vaccination. Almost all (98.0%) of the study participants indicated that, to protect the public, health professionals should follow government guidelines about vaccines.

Table 5 presents results on summary statistics of the computed composite scores on perceptions and attitudes of the HCPs toward the COVID-19 vaccination. The overall median score (with IQR) of perception on COVID-19 vaccination was positive [6.00; IQR = 4.00–7.00] out of a maximum of 10.00. This was scored by 61.5% of the health professionals. The overall median score (with IQR) of attitudes towards the COVID-19 vaccination was positive [5.00; IQR = 4.00–5.00] out of a maximum of 6, scored by 50.6% of the HCPs. The data distribution for both perceptions and attitudes towards the COVID-19 vaccination was negatively skewed and leptokurtic.

## Predictors of health professionals' attitudes towards the COVID-19 vaccination

Table 6 presents the results of the predictors of health professionals' attitudes towards the COVID-19 vaccination. As presented in Table 6, positive perception (ρ < 0.05), female sex (ρ < 0.05), marital status (ρ < 0.01), having a bachelor's degree or more (ρ < 0.05), and working in Wa North sub-municipal (ρ < 0.05) were statistically significantly associated with positive attitudes towards COVID-19 vaccination. Health professionals who had a positive perception compared to those with a negative perception of COVID-19 vaccination had higher odds of

**Table 3. Health professionals' perceptions on COVID-19 vaccination (n = 403).**

| Perception about COVID-19 Vaccines | Frequency (*n* = 403) | Percentage (%) |
|---|---|---|
| **Knows infected persons** | | |
| No | 304 | 75.4 |
| Yes | 99 | 24.6 |
| **Contact with infected persons** | | |
| No | 308 | 76.4 |
| Yes | 95 | 23.6 |
| **Fear of getting infected** | | |
| No | 127 | 31.5 |
| Yes | 276 | 68.5 |
| **Reality of COVID-19** | | |
| No | 6 | 1.5 |
| Yes | 397 | 98.5 |
| **Protective measures at work place** | | |
| No | 5 | 1.2 |
| Yes | 398 | 98.8 |
| **Safest Vaccines** | | |
| Oxford Astra-Zeneca vaccines | 84 | 20.8 |
| Gamaleya Sputnik V | 15 | 3.7 |
| Pfizer/BioNtech Comirnaty | 124 | 30.8 |
| Janssen (Johnson & Johnson) | 102 | 25.3 |
| Serum institute of India Covishield (Oxford /Astra-Zeneca formulation | 37 | 9.2 |
| All Vaccines | 238 | 59.1 |
| None | 7 | 1.7 |
| **Vaccination prevents spread of infection among patients** | | |
| Agreed | 295 | 73.2 |
| Disagreed | 14 | 3.5 |
| Neutral | 94 | 23.3 |
| **Vaccination prevents infection among hospital workers** | | |
| Agreed | 290 | 72.0 |
| Disagreed | 19 | 4.7 |
| Neutral | 94 | 23.3 |
| **Vaccines are safe and effective** | | |
| Agreed | 285 | 70.7 |
| Disagreed | 12 | 3.0 |
| Neutral | 106 | 26.3 |
| **The vaccines are immune-potent** | | |
| Agreed | 238 | 59.1 |
| Disagreed | 20 | 5.0 |
| Neutral | 145 | 36.0 |

exhibiting positive attitudes towards COVID-19 vaccination [aOR = 1.81; 95% CI = 1.14–2.87]. Female health professionals compared to their male counterparts were 42% less likely to accept COVID-19 vaccinations [aOR = 0.58; 95% CI = 0.35–0.97]. Health professionals who were married compared to those who were single were about 2 times more likely [aOR = 1.94; 95% CI = 1.20–3.12] to possess positive attitudes towards COVID-19 vaccination. Health professionals who had at least a bachelor's degree compared to those with certificates were about 2 times more likely [aOR = 2.03, 95% CI = 1.0–11.12] to exhibit positive attitudes towards

**Table 4. Attitudes towards COVID-19 vaccination (n = 403).**

| Attitude variables of participants | Frequency | Percentage (%) |
|---|---|---|
| **Ever had a COVID-19 Infection** | | |
| No | 388 | 96.3 |
| Yes | 15 | 3.7 |
| **Ever received COVID-19 Vaccination** | | |
| No | 32 | 7.9 |
| Yes | 371 | 92.1 |
| **Type of Vaccine received** | | |
| Moderna Spikevax | 35 | 8.7 |
| Oxford Astra-Zeneca | 154 | 38.2 |
| Gamaleya Sputnik V | 25 | 6.2 |
| Pfizer/BioNtech Comirnaty | 78 | 19.4 |
| Janssen (Johnson & Johnson) | 48 | 11.9 |
| Serum institute of India Covishield (Oxford /Astra-Zeneca formulation | 89 | 22.1 |
| **Completed COVID-19 vaccination** | | |
| No | 49 | 12.2 |
| Yes | 322 | 79.9 |
| Not applicable | 32 | 7.9 |
| **Reasons for not completing Vaccination** | | |
| Side Effects of initial shot (s) | 47 | 11.6 |
| I'm not sure of safety of vaccine | 1 | 0.2 |
| Yet to take my last dose | 1 | 0.2 |
| Not applicable | 354 | 87.8 |
| **Reason(s) for not taking COVID-19 vaccines** | | |
| Infertility | 10 | 2.5 |
| Severe Allergic Reactions | 1 | 0.2 |
| Unknown long-term effects | 26 | 6.5 |
| Not applicable | 366 | 90.8 |
| **Receipt of COVID-19 vaccine in future** | | |
| No | 10 | 2.5 |
| Yes | 22 | 5.5 |
| Not applicable | 371 | 92.1 |
| **Alternative COVID-19 preventive measure** | | |
| Covid-19 prevention protocol | 359 | 89.1 |
| Physical Exercise | 24 | 6.0 |
| Praying | 18 | 4.5 |
| Traditional Medicine | 2 | 0.5 |
| **Prioritising chronically/severely ill for vaccination** | | |
| No | 93 | 23.1 |
| Yes | 310 | 76.9 |
| **Prioritising Healthcare workers for vaccination** | | |
| No | 59 | 14.6 |
| Yes | 344 | 85.4 |
| **Adherence to government guidelines about vaccines** | | |
| No | 8 | 2.0 |
| Yes | 395 | 98.0 |

**Table 5. Composite distribution of healthcare professionals' perceptions and attitudes towards COVID-19 vaccination.**

| Variables | Frequency (%) |
| --- | --- |
| **Perception on COVID-19 vaccination** | |
| Negative | 38.5% |
| Positive | 61.5% |
| Mean (SD) | 5.41 (± 1.70) |
| Median (IQR) | 6.00 (4.00–7.00) |
| Minimum | 1.00 |
| Maximum | 7.00 |
| Range | 6.00 |
| Skewness | -0.791 |
| Kurtosis | 2.28 |
| **Attitudes towards COVID-19 vaccination** | |
| Poor | 49.4% |
| Good | 50.6% |
| Mean (SD) | 4.37 (± 0.76) |
| Median (IQR) | 5.00 (4.00–5.00) |
| Minimum | 1.00 |
| Maximum | 5.00 |
| Range | 4.00 |
| Skewness | -1.40 |
| Kurtosis | 5.44 |

SD = Standard deviation; IQR = Inter quartile range

COVID-19 vaccination. Wa North sub-municipal health professionals compared to those within the Bamaha sub-municipal were 78% less likely [aOR = 0.22; 95% CI = 0.05–0.96] to exhibit good attitudes towards COVID-19 vaccination.

## Discussions

### Perceptions and attitudes of healthcare professionals towards COVID-19 vaccination

Our study assessed HCPs' perceptions and the predictors of their attitudes towards the COVID-19 vaccination in the Wa Municipality, Upper West Region, Ghana. The present study reported a positive perception of COVID-19 vaccination among HCPs. This perception level was higher than the findings from studies conducted among health workers in Ethiopia [9] and Nigeria [17]. Positive perceptions of COVID-19 vaccination among health professionals may be linked to improvements in the strategies for COVID-19 infection prevention and control. These strategies encompass improved awareness and knowledge of COVID-19, improved testing capacity [18], improved treatment [19], scientific advancement [20], readily available and accessible vaccines, and vaccination efforts [1]. Therefore, a positive perception of the COVID-19 vaccine portends an increased acceptance of the vaccine among HCPs as well as the general population. Improved COVID-19 vaccination rates would contribute to achieving policy-targeted herd immunity and ultimately reduce severe COVID-19 infections. Given this finding, healthcare policymakers need to design interventions that will improve communication, foster positive perception, and ultimately sustain and improve COVID-19 vaccinations.

**Table 6. Predictors of attitudes of health professionals towards COVID-19 vaccination.**

| Variable | Model 0 | Model 1 | Model 2 | Model 3 |
|---|---|---|---|---|
| | aOR [95% CI] | aOR [95% CI] | aOR [95% CI] | aOR [95% CI] |
| **Fixed effects** | | | | |
| **Perception of COVID-19** | | | | |
| Negative | | Ref | Ref | Ref |
| Positive | | 1.69* [1.10–2.61] | 2.20***[1.42–3.39] | 1.81* [1.14–2.87] |
| **Sex** | | | | |
| Male | | Ref | | Ref |
| Female | | 0.49** [0.31–0.79] | | 0.58* [0.35–0.97] |
| **Marital status** | | | | |
| Single | | Ref | | Ref |
| Married | | 1.84** [1.17–2.91] | | 1.94**[1.20–3.12] |
| **Education Qualification** | | | | |
| Certificate | | Ref | | Ref |
| Diploma | | 0.86 [0.54–1.37] | | 1.0 [0.61–1.63] |
| Degree/ above | | 1.89 [0.96–3.73] | | 2.03* [1.01–4.12] |
| **Sub-Municipal** | | | | |
| Bamahu | | | Ref | Ref |
| Busa | | | 0.43 [0.05–3.56] | 0.58 [0.06–5.73] |
| Charia | | | 0.39 [0.05–2.99] | 0.36 [0.05–2.79] |
| Charingu | | | 0.57 [0.04–8.13] | 0.56 [0.04–8.74] |
| Dobile | | | 0.46 [0.12–1.80] | 0.50 [0.12–2.05] |
| Kambali | | | 0.56 [0.14–2.29] | 0.58 [0.14–2.43] |
| Kperisi | | | 0.11 [0.01–1.14] | 0.12 [0.01–1.18] |
| Kpongu | | | 1.32 [0.18–9.67] | 1.60 [0.22–11.65] |
| Wa North | | | 0.18* [0.04–0.73] | 0.22* [0.05–0.96] |
| Wa South | | | 0.40 [0.10–1.57] | 0.44 [0.11–1.81] |
| **Facility location** | | | | |
| Rural | | | Ref | Ref |
| Urban | | | 0.61 [0.2–1.86] | 0.57 [0.19–1.68] |
| **Random effects and model fitness** | | | | |
| Constant | 1.34**[1.10–1.64] | 1.01* [0.57–1.80] | 3.22* [0.69–14.94] | 2.85* [0.60–13.6] |
| Obs | 403 | 403 | 403 | 403 |
| Prob > chi2 | | $p < 0.0001$ | $p = 0.0009$ | $p < 0.0001$ |
| Pseudo R2 | | 0.0675 | 0.0627 | 0.1014 |
| Log pseudo-Likelihood | -275.0 | -256.4 | -257.8 | -247.1 |
| Wald χ2 | Ref | χ2 (5) = 32.12 | χ2 (11) = 31.47 | χ2 (15) = 48.14 |
| AIC | 552.0078 | 524.8702 | 539.5052 | 526.2525 |

aOR = Adjusted Odds Ratio; CI = Confidence Interval; AIC = Akaike's Information Criterion

*ρ < 0.05

**ρ < 0.01

***p < 0.001

Model 0 = Null; Model 1 = perception, individual level factors; Model 2 = perception and contextual factors; Model 3 = perception, individual level and contextual factors

Despite the overall positive perception of the COVID-19 vaccination among health professionals, there were several important gaps. About a quarter of the HCPs did not perceive that the COVID-19 vaccine was effective, safe, and prevented infection among patients and health workers alike. Also, nearly half of the health professionals did not perceive that the COVID-19 vaccines were immune-potent. These gaps, if not addressed, have the potential to fuel negative perceptions of the COVID-19 vaccination among these health professionals. However, Abduljaleel [10] reported a negative perception among HCPs in Kano State, Nigeria. The difference in findings may be explained in terms of the methods and sample size of the study. Abduljaleel [10] recruited 1004 participants in a mixed-method study, whereas our study and those of Adane et al. [9] and Adejumo et al. [17] used purely quantitative methods.

Findings from this current study revealed that slightly more than half (50.6%) of health professionals in Wa Municipal had a good attitude toward the COVID-19 vaccines, which is not impressive considering the fact that the majority had a positive perception. Similar findings of a good attitude towards COVID-19 vaccination among HCPs have been reported by Tolossa et al. [15] and Adane et al. [9] in Ethiopia. Kabamba et al. [21] in the Democratic Republic of Congo reported a higher rate (67.4%) of good attitudes towards COVID-19 vaccination among health professionals, while Tharwat et al. [22] in Egypt and Jankowska-Polańska et al. [23] in Poland even reported a higher rate (77.4%) of attitudes towards COVID-19 vaccination. As postulated by the health belief model, positive attitudes towards COVID-19 vaccination among the HCPs may be related to their increased perceived risk, susceptibility, and threat of COVID-19 infection. Additionally, as the PICA-A framework theorized, the health professionals' positive attitudes towards the COVID-19 vaccination could influence their decision to accept the COVID-19 vaccine.

In contrast to our findings, Agyekum et al. [24] reported negative attitudes towards COVID-19 vaccination among HCPs in Ghana. As evident in our study, these negative attitudes towards COVID-19 vaccination were largely attributable to the perceptions of health professionals regarding vaccine safety and side effects. In our study, 49% of the respondents did not complete their vaccination schedule due to concerns about vaccine safety and side effects. Agyekum et al. [24] reported that 65.5% of the health professionals had concerns about the safety of the vaccine, while 14.8% of them were concerned about the side effects of the vaccine. These differences in findings may be attributable to differences in sample size and sample population. Jankowska-Polańska et al.'s [23] study involved largely mainstream healthcare workers and medical students, while the other studies (including our study) sampled solely HCPs. Additionally, Agyekum et al. [24] in particular used a smaller sample size. Policymakers can help improve acceptance of the COVID-19 vaccination by addressing specific attitude-related problems among healthcare professionals in the Wa Municipality.

## Predictors of healthcare professionals' attitudes towards COVID-19 vaccination

The perceptions of the HCPs were linked to their attitudes towards the COVID-19 vaccinations. In consonance with our findings are the findings of Jing et al. [25] in China, Adane et al. [9] in Ethiopia, and Amponsah-Tabi et al. [14] in Ghana. Positive perceptions of the COVID-19 vaccination among the HCPs imply a good impression of the role of the COVID-19 vaccine. This good impression plausibly influences healthcare professionals' attitudes towards the COVID-19 vaccination. Therefore, our finding confirms the PICA-A framework presented in this study.

The female health professionals were statistically significantly less likely to exhibit good attitudes towards the COVID-19 vaccination. Shekhar et al. [26] in the USA, Ledda et al. [27] in

Italy, Ahmed et al. [28] in Ethiopia, and Agyekum et al. [24] in Ghana reported similar findings. Therefore, contrary to the popular belief that females have better health-seeking behavior compared to males, our findings demonstrate otherwise. This can be explained in terms of risk perception and an affinity for adventure. Generally, men compared to females have poorer health and, hence, may tend to perceive themselves as being at greater risk of infection. Additionally, since the COVID-19 vaccine is novel and has been associated with uncertainties, the female health professionals were probably risk-averse and probably not adventurous enough to entertain the risk of accepting the COVID-19 vaccine. In line with the postulations of the health belief model and the PICA-A framework, perceptions of risk influence attitudes and acceptance of the COVID-19 vaccine. Therefore, targeted education to demystify the uncertainties surrounding the COVID-19 vaccine should be emphasized, especially among female health workers.

Married health professionals were statistically significantly more likely to show positive attitudes towards the COVID-19 vaccination. Consistent with our findings are the findings of Wang et al. [29] in China and Ahmed et al. [28] in Ethiopia. Marriage confers an acceptable degree of awareness, risk perceptions, and collective responsibility, which improves health-seeking behavior [30]. Married health professionals, plausibly in an attempt to protect themselves and hence their families from being infected, may be perceived to be relatively at higher risk of infection on the one hand. On the other hand, married people may have improved awareness of the search for a better family life [30]. However, the findings of Rahman et al. [31] in Bangladesh and Aklil et al. [32] in Ethiopia were different. This difference is explainable in terms of the difference in target populations sampled. Rahman et al. [31] sampled the general Bangladeshi adult population, while Aklil et al. [32] studied Northwestern Ethiopian college students.

Health professionals who had a bachelor's degree and above, compared to those with a certificate as their highest educational qualification, were more likely to exhibit positive attitudes towards COVID-19 vaccinations. The findings of Paul et al. [33] in the UK, Ledda et al. [27] in Italy, Ahmed et al. [28] in Ethiopia, and Adejumo et al. [17] in Nigeria among healthcare workers corroborated our findings. Higher educational levels are associated with increased awareness, knowledge, and access to information [30]. Therefore, healthcare workers who have higher educational qualifications plausibly have a greater advantage in terms of access to available information about the COVID-19 vaccination [30]. Additionally, well-informed persons in this context, including health care professionals, may perceive themselves to be at higher risk of infection and are hence more likely to adopt positive attitudes towards the COVID-19 vaccination. However, Wang et al. [29] reported no statistically significant association between educational level and attitudes towards COVID-19 vaccination. This discrepancy may be explained in terms of the relative availability of COVID-19 vaccination-specific information. As of the time Wang et al. [29] conducted their study, SARS-CoV 2 was novel, and therefore, there was insufficient information regarding the infection as well as the prospects of vaccination. This phenomenon of insufficient information prevailed at the time, regardless of level of education.

HCPs in the Wa North sub-municipal were 78% less likely to exhibit good attitudes towards COVID-19 vaccination as compared to those within the Bamaha sub-municipal. This finding may be explained in terms of the contextual setting and risk perception. Wa North sub-municipal is relatively more rural compared to Bamahu sub-municipal and hence may not have been perceived as being at high risk of infection, as the disease burden was mainly in the relatively more urban locations. Additionally, a deductive assessment of the data revealed that a greater proportion of the health professionals in Wa North sub-municipal (42.6%) compared to those in Bamahu (33.3%) had a negative perception of the COVID-19 vaccination. According to the

PICA-A framework, this negative perception could have influenced the lower odds of Wa North sub-municipal HCPs exhibiting positive attitudes. There is therefore a need for targeted education for rural dwellers who may not be privy to important information concerning COVID-19.

## Strengths and limitations of the study

Our study had strengths and limitations. Our study is novel in that it sought to assess the links among perception, individual, and contextual predictors of attitudes towards COVID-19 vaccination. We administered a valid and reliable data collection instrument to participants who were sampled via a probabilistic sampling technique (multistage sampling technique). Additionally, we used rigorous hierarchical logit modeling to eliminate spurious and confounding factors that could affect the predictors of attitudes towards COVID-19 vaccination. Although a representative sampling strategy was used, the complete representativeness of the sample cannot be guaranteed. Our study, which sampled only HCPs in Wa Municipality, relied on self-reported responses. Consequently, our study may be prone to recall and social desirability biases. We therefore recommend that future studies explore the links among perception, individual, contextual factors, and attitudes towards COVID-19 vaccination using a more representative multi-regional or national study.

## Conclusion

The HCPs' perceptions and attitudes towards the COVID-19 vaccination were positive but suboptimal. The healthcare professionals' perception of COVID-19, sex, marital status, educational status, and sub-municipality predicted their attitudes towards the COVID-19 vaccination. Regular education on COVID-19 vaccine benefits, safety, efficacy, providing a supportive work environment, and addressing vaccine availability and accessibility for HCPs is recommended. These interventions should particularly focus on female, single HCPs who possess a below-bachelor's degree and are working in the Wa North sub-municipal area.

## Supporting information

**S1 Questionnaire. Perception and attitudes towards COVID-19 vaccination.**
(DOCX)

**S1 Data. Minimum dataset from survey.**
(DTA)

## Author Contributions

**Conceptualization:** Augustine Ngmenemandel Balegha.

**Data curation:** Augustine Ngmenemandel Balegha.

**Formal analysis:** Augustine Ngmenemandel Balegha, Louis Mornah.

**Investigation:** Augustine Ngmenemandel Balegha, Suburu Abdul-Aziz, Louis Mornah.

**Methodology:** Augustine Ngmenemandel Balegha.

**Project administration:** Suburu Abdul-Aziz, Louis Mornah.

**Resources:** Suburu Abdul-Aziz.

**Supervision:** Suburu Abdul-Aziz.

**Validation:** Augustine Ngmenemandel Balegha.

**Writing – original draft:** Augustine Ngmenemandel Balegha.

**Writing – review & editing:** Augustine Ngmenemandel Balegha, Suburu Abdul-Aziz, Louis Mornah.

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
