## [Decision Letter · Decision Letter 0]

11 Jan 2024

PONE-D-23-35761Perception and COVID-19 vaccination attitudes among healthcare professionals: a multi-centre cross-sectional hierarchical logit analysis in North-Western GhanaPLOS ONE

Dear Dr. Balegha, Thank you for submitting your manuscript to PLOS ONE. After careful consideration, we feel that it has merit but does not fully meet PLOS ONE’s publication criteria as it currently stands. Therefore, we invite you to submit a revised version of the manuscript that addresses the points raised during the review process.

We look forward to receiving your revised manuscript.

Kind regards,

Pracheth Raghuveer, MD, DNB

Academic Editor

PLOS ONE

Journal Requirements:

Reviewers' comments:

Reviewer's Responses to Questions

**Comments to the Author**

1. Is the manuscript technically sound, and do the data support the conclusions?

Reviewer #1: Yes

Reviewer #2: Yes

2. Has the statistical analysis been performed appropriately and rigorously? 

Reviewer #1: Yes

Reviewer #2: Yes

3. Have the authors made all data underlying the findings in their manuscript fully available?

Reviewer #1: Yes

Reviewer #2: Yes

4. Is the manuscript presented in an intelligible fashion and written in standard English?

Reviewer #1: Yes

Reviewer #2: Yes

5. Review Comments to the Author

Reviewer #1: I am pleased to review and provide my opinion on the topic titled 'Perception and COVID-19 Vaccination Attitudes Among Healthcare Professionals: A Multi-Centre Cross-Sectional Hierarchical Logit Analysis in North-Western Ghana.'

Overall, the article provides a comprehensive analysis of healthcare professionals' perceptions and attitudes toward COVID-19 vaccination in the Wa Municipality, Upper West Region, Ghana. However, there are some areas that could be improved for clarity and precision:

• The current title is broad. Consider enhancing specificity and conciseness to better reflect the study's main focus. A suggested revised title is: 'Healthcare Professionals' Perception and COVID-19 Vaccination Attitudes in North-Western Ghana: A Multi-Centre Analysis.'

• The abstract should provide a concise summary of the study, including the objectives, methods, results, and conclusion. It currently lacks a clear statement of the study's purpose.

• The introduction should clearly state the problem, the significance of the study, and the research questions or objectives.

• It's important to include a literature review to provide context and highlight the gap in knowledge that this study aims to fill.

• I would suggest you seen how to incorporate the following systematic review https://journals.plos.org/plosone/article?id=10.1371/journal.pone.0289295

• Provide context for the problem: What is the actual issue in the Wa Municipality, Upper West Region?

• Clarify the sampling method used to select the 420 health professionals initially sampled.

• Offer additional information about the survey tool, including its development or adaptation. However, the reliability (Cronbach's alpha = 0.67, α = 0.64, and α = 0.60) and validity appear questionable. Was the correct data collected?"

• Some of the tables have a lot of information; consider breaking them down for easier interpretation.

• Use clear and concise language in presenting the results to enhance readability.

• Provide a more detailed discussion of the results, relating them back to the existing literature.

• You don’t need sub-headings the discussion section

• Discuss the implications of the findings for public health and healthcare policy.

• Table 2: Consider providing a clearer title that indicates it is presenting sociodemographic characteristics.

• Table 3: Use more descriptive headers for each column to improve clarity.

• Table 4: Consider organizing the data in a more structured format for better readability.

• Table 5: Consider providing more context for the computed composite scores.

• Check for grammatical errors and ensure consistency in writing style.

• Avoid redundancy and ensure each section adds unique information to the study.

• Provide a concise summary of the key findings and their significance.

• Suggest practical implications and recommendations based on the study.

Reviewer #2: Dear authors and editors,

Thank you for the invitation to review this manuscript entitled “Perception and COVID-19 vaccination attitudes among healthcare professionals: a multi-centre cross-sectional hierarchical logit analysis in North-Western Ghana”.

Overall is an interesting and relevant manuscript, however, there are some points that the authors need to clarify and address for further improvement.

Lines 146-147, the sample size was estimated based on the proportion of the COVID-19 vaccination rate among the HCPs = 0.5. This study aimed to investigate the perception of healthcare professionals, and the sample size is based on the vaccination rate. It would be nice if the authors explained the justification for applying the vaccination rate to estimate the sample size.

Line 192: the authors mentioned that content validity was examined by three consultants. Kindly include the content validity index of the items (if analyzed).

Line 197: The authors mentioned that “construct validity of the instrument using inter-item correlation”. Inter-item correlation is usually applied for the reliability analysis. Kindly include the other construct validity such as exploratory factor analysis and confirmatory factor analysis findings.

Kindly prepare the tables in three-line format.

Line 321: Table 5 might not need to be included in the main manuscript. If the authors would like to report, it would be better to include it as an appendix table.

Thank you.

6. PLOS authors have the option to publish the peer review history of their article (what does this mean?). If published, this will include your full peer review and any attached files.

Reviewer #1: **Yes: **Dr Amir Kabunga

Reviewer #2: No

---

## [Author Response · Author response to Decision Letter 0]

21 Jan 2024

POINT-BY-POINT RESPONSE TO THE COMMENTS 

Ref: Submission ID: [PONE-D-23-35761] - [EMID:7ca4e78db36560bb]

Title: "Perception and COVID-19 vaccination attitudes among healthcare professionals: a

multi-centre cross-sectional hierarchical logit analysis in North-Western Ghana"

Journal Requirements:

and 

Response: We have checked our manuscript to ensure it meets PLOS ONE’s style requirement including those for file naming.

Response: We employed the services of a colleague who edited our manuscript.

i. The name of the colleague or the details of the professional service that edited your manuscript

Response: Mr. David Wuollah-Dire (PhD Student), edited our manuscript.

ii. A copy of your manuscript showing your changes by either highlighting them or using track changes (uploaded as a *supporting information* file).

Response: We have labeled a copy of the manuscript with changes 'Revised Manuscript with Track Changes' and have uploaded same as supporting information file

iii. A clean copy of the edited manuscript (uploaded as the new *manuscript* file)”

Response: We have uploaded a clean copy of the edited manuscript as new manuscript file

Response: We have reviewed our references for completeness and correctness.

Reviewers’ Comments: 

Reviewer 1 

General comment: I am pleased to review and provide my opinion on the topic titled 'Perception and COVID-19 Vaccination Attitudes Among Healthcare Professionals: A Multi-Centre Cross-Sectional Hierarchical Logit Analysis in North-Western Ghana.' 

Overall, the article provides a comprehensive analysis of healthcare professionals' perceptions and attitudes toward COVID-19 vaccination in the Wa Municipality, Upper West Region, Ghana. However, there are some areas that could be improved for clarity and precision:

Response: We are very grateful to the reviewer for the positive comments made concerning our manuscript. We are equally grateful for the useful critical comments made, of which we have painstakingly addressed. We hope that our responses to your comments have improved the quality of the manuscript in terms of its suitability for publication. Below is a point-to-point response to the specific comments of the reviewer.

1. The current title is broad. Consider enhancing specificity and conciseness to better reflect the study's main focus. A suggested revised title is: 'Healthcare Professionals' Perception and COVID-19 Vaccination Attitudes in North-Western Ghana: A Multi-Centre Analysis.' 

Response: We are grateful for the comment. We quite agree with your suggestion and have revised the title to read “Healthcare professionals' perception and COVID-19 vaccination attitudes in North-Western Ghana: a multi-center analysis”.

2. The abstract should provide a concise summary of the study, including the objectives, methods, results, and conclusion. It currently lacks a clear statement of the study's purpose. 

Response: Thank you for the comment. The abstract is already structured with sub-sections- introduction, methods, results and conclusion. We have also revised the introduction sub-section of the abstract to include a clear statement of the study’s purpose and objectives as indicated below (please see page 1: lines 18-21).

“However, the perceptions and attitudes of HCPs towards COVID-19 vaccination remain largely unexplored. We therefore assessed healthcare professionals’ perceptions, attitudes and predictors of their attitudes towards COVID-19 vaccination in the Wa Municipality, Upper West Region of Ghana.”

3. The introduction should clearly state the problem, the significance of the study, and the research questions or objectives 

Response: We have now clearly stated the problem of the study (please see page 4, lines 84-86)

Therefore, a knowledge gap exists in terms of healthcare professionals’ perception and attitudes towards COVID-19 vaccination in the Wa Municipality, Upper West Region of Ghana.

The objectives of the study have also been clearly stated to read:

“Therefore, this study assessed the perceptions, attitudes and predictors of attitudes towards COVID-19 vaccination among HCPs in the Wa Municipality of the UWR of Ghana. Such research is crucial in designing effective vaccination campaigns and addressing concerns or barriers to HCPs’ perceptions and attitudes towards COVID-19 vaccination.” Please see page 4; lines 87 & 88.

The significance of the study has already been stated, as shown below: 

Such research is crucial in designing effective vaccination campaigns and addressing concerns or barriers to HCPs’ perceptions and attitudes towards COVID-19 vaccination. (please see page 4: lines 88-90).

4. It's important to include a literature review to provide context and highlight the gap in knowledge that this study aims to fill. 

Response: We are pleased to state that in line with PLOS ONE’s journal formatting guidelines, we did not need to write a literature review as a stand-alone subsection. However, in line with your suggestion, we have reviewed relevant literature in the introduction subsection which contextualizes and highlights knowledge gaps in the study. Please see below: 

Several studies have examined HCPs' attitudes and perceptions towards the COVID-19 vaccination globally.

In a Bangladeshi cross-sectional community survey, Islam et al. [6] reported positive attitudes towards the COVID-19 vaccination. Similarly, a public survey in Jordan reported positive attitudes towards the COVID-19 vaccination [7]. Likewise, studies conducted in sub-Saharan Africa portray mixed findings. In Ethiopia, Adane et al. [8] reported that the HCPs had a good perception (60.5%) and positive attitudes (52.3%) towards the COVID-19 vaccination. In Nigeria, Abduljaleel et al. [9] reported a poor perception (24.3%) of COVID-19 vaccination among nurses. In Ghana, there is a paucity of studies examining the perceptions and attitudes of HCPs towards COVID-19 vaccinations. These few studies have largely concentrated on COVID-19 vaccine acceptance among HCPs and not their overall attitudes towards COVID-19 vaccination. Additionally, no study linking perceptions, individual-level factors, and contextual factors to the attitudes of HCPs has been conducted. Moreover, studies on COVID-19 in Ghana have mostly focused on southern Ghanaian cities, to the neglect of relatively resource-poor regions like the Upper West Region (UWR) of Ghana. (please see page 4; lines 70-84)

5. I would suggest you seen how to incorporate the following systematic review https://journals.plos.org/plosone/article?id=10.1371/journal.pone.0289295

Response: We have incorporated the suggested systematic review by Kigongo et al. (2023) into our manuscript. Please see pages 3 and 4; lines 59-61 and 68-71.

6. Provide context for the problem: What is the actual issue in the Wa Municipality, Upper West Region? 

Response: Context for the problem has now been provided, as indicated below: 

Therefore, a knowledge gap exists in terms of healthcare professionals’ perception and attitudes towards COVID-19 vaccination in the Wa Municipality, Upper West Region of Ghana. Please see page 4: lines 84-86.

7. Clarify the sampling method used to select the 420 health professionals initially sampled 

Response: Multistage cluster sampling technique was used to sample the healthcare professionals. Please refer to page 7: line 157.

8. Offer additional information about the survey tool, including its development or adaptation. However, the reliability (Cronbach's alpha = 0.67, α = 0.64, and α = 0.60) and validity appear questionable. Was the correct data collected?" 

Response: We are very grateful for your comment. The questionnaire development and adaptation has been stated under the subsection data collection techniques and tools, which is part of the methods section (Please see pages 8 & 9: lines 175-194).

The reliability was computed using Cronbach alpha while the validity was tested using face validity, content validity (Content validity index now computed) and construct validity (comparative factor analysis has been added). The tool was used to collect data on the socio-demographics (both individual and contextual factors), perception and attitudes of the healthcare professionals. Please see page 9: lines 195-209

9. Some of the tables have a lot of information; consider breaking them down for easier interpretation. 

Response: The tables have been presented in line with APA 7 requirement. We have revised the headings of some of the tables to ensure specificity and clarity, as may have been addressed in the ensuing questions.

10. Use clear and concise language in presenting the results to enhance readability 

Response: We have proofread the entire manuscript, including the results section using clear and concise language. Therefore, we believe and hope that, the manuscript is now readable enough.

11. Provide a more detailed discussion of the results, relating them back to the existing literature. 

Response: To the best of our knowledge, each relevant finding of the study has been discussed in detail. We have endeavored to state the practical implications of the study findings and where available, we have cited and linked the results to studies with similar findings as well as studies with contrasting findings. We have also offered plausible explanations as to why our results vary from other studies. Theoretical, policy and public health implications of the study’s findings have also been stated, where applicable. 

12. You don’t need sub-headings the discussion section 

Response: We are grateful for you comment. We quite recognize the possibility of writing the discussion section without sub-headings. However, the journal’s prescription of manuscript organization permits the use of subheadings in the discussion section. Therefore, we decided to subdivide the discussion section.

13. Discuss the implications of the findings for public health and healthcare policy 

Response: Thank you for the comment. We are pleased to indicate that although the public health and healthcare policy implications were not written under a distinct sub-heading, we endeavored to include it as part of the discussion of the important findings. We have added the following as you have suggested: 

Given this finding, healthcare policymakers need to design interventions that will improve communication, foster positive perception, and ultimately sustain and improve COVID-19 vaccinations. (please see page 22; lines 358-362)

Policymakers can help improve acceptance of the COVID-19 vaccination by addressing specific attitude-related problems among healthcare professionals in the Wa Municipality. (page 24; lines 397-399)

Therefore, targeted education to demystify the uncertainties surrounding the COVID-19 vaccine should be emphasized, especially among female health workers. (was available in the original manuscript. Please see page 25; lines 419-421)

There is therefore a need for targeted education for rural dwellers who may not be privy to important information concerning COVID-19. (please see page 26; lines 460-462)

14. Table 2: Consider providing a clearer title that indicates it is presenting sociodemographic characteristics 

Response: We have revised the heading of Table two (please see below)

Table 2: Socio-demographic characteristics and healthcare professionals’ attitudes towards COVID-19 vaccination (n=403)

15. Table 3: Use more descriptive headers for each column to improve clarity 

Response: We are grateful for your comment. To the best of our knowledge, the header for the first column describes each sub-variable under the main variable, perception of healthcare professionals. The second column’s header represents the frequencies in terms of numbers while the last header represents the percentage frequency of occurrence. 

16. Table 4: Consider organizing the data in a more structured format for better readability. 

Response: We have now revised the tables in accordance with the APA guidelines (APA 7), using the three-line format

17. Table 5: Consider providing more context for the computed composite scores 

Response: We have now revised the title of Table 5 to provide more context (please see below):

Table 5: Composite distribution of healthcare professionals’ perceptions and attitudes towards COVID-19 vaccination

18. Check for grammatical errors and ensure consistency in writing style. 

Response: We have copyedited the work and we hope that we have addressed the issues relating to grammatical errors and writing style.

19. Avoid redundancy and ensure each section adds unique information to the study. 

Response: We have proofread the manuscript and we believe to the best of our knowledge we have now eliminated redundant statements

20. Provide a concise summary of the key findings and their significance. 

Response: The key findings and their significance have been presented in the conclusion section. We have revised the conclusion section to include predictors of health professionals’ attitudes towards COVID-19 vaccination (emboldened below). The conclusion section now reads:

The HCPs’ perceptions and attitudes towards the COVID-19 vaccination were positive but suboptimal. The healthcare professionals’ perception of COVID-19, sex, marital status, educational status and sub-municipality predicted their attitudes towards COVID-19 vaccination. Regular education on COVID-19 vaccine benefits, safety, efficacy, providing a supportive work environment, and addressing vaccine availability and accessibility for HCPs is recommended. These interventions should particularly focus on female, single HCPs who possess a below-bachelor’s degree and are working in the Wa North sub-municipal area.

21. Suggest practical implications and recommendations based on the study. 

Response: The practical implications of the findings have been included in the discussion of the results (pages 22-27; lines 346-475). Recommendations based on the study’s results have also been included both in the discussion section and under conclusion section (page 26: lines 469-472)

Reviewer #2: Dear authors and editors,

Thank you for the invitation to review this manuscript entitled “Perception and COVID-19 vaccination attitudes among healthcare professionals: a multi-centre cross-sectional hierarchical logit analysis in North-Western Ghana”.

Overall is an interesting and relevant manuscript, however, there are some points that the authors need to clarify and address for further improvement. Thank you for the invitation to review this manuscript entitled “Perception and COVID-19 vaccination attitudes among healthcare professionals: a multi-centre cross-sectional hierarchical logit analysis in North-Western Ghana”.

Response: We are very grateful for this profound commendation concerning our manuscript. Having addressed your comments, we hope that the quality of our manuscript has improved, and that our manuscript is now suitable for publication

Specific comments

 Lines 146-147, the sample size was estimated based on the proportion of the COVID-19 vaccination rate among the HCPs = 0.5. This study aimed to investigate the perception of healthcare professionals, and the sample size is based on the vaccination rate. It would be nice if the authors explained the justification for applying the vaccination rate to estimate the sample size.

Response: We are very grateful for the comment. We actually intended to use an indicator of healthcare professional’s attitudes toward COVID-19 vaccination; for example, proportion of Ghanaian healthcare professionals with good attitudes towards COVID-19. We have therefore revised the statement to read:

“We used Cochran’s formula to estimate the sample size of the study. According to Cochran [13], the sample size of a population may be estimated using: n=(z^2 pq)/d^2 ; where n is the sample size, z =1.96 at 95% confidence level, p = the proportion of healthcare professionals with good attitudes towards the COVID-19 vaccination = 0.5 (largest assumed estimate for p), q = 1-p, and d = estimated margin of error = 0.05.” (please see lines 146-151)

 Line 192: the authors mentioned that content validity was examined by three consultants. Kindly include the content validity index of the items (if analyzed).

Response: The content validity index (CVI) has now been stated as 0.78. Please see page 9; lines 196 & 197

 Line 197: The authors mentioned that “construct validity of the instrument using inter-item correlation”. Inter-item correlation is usually applied for the reliability analysis. Kindly include the other construct validity such as exploratory factor analysis and confirmatory factor analysis findings.

Response: We are very grateful for this insightful comment. We have deleted the statement on inter-item correlation being used for testing construct validity. We have done a confirmatory factor analysis and have reported its relevant indices on page 9; lines 201-205. We have therefore replaced these statements “We then determined the construct validity of the instrument using inter-item correlation. The average total inter-item correlation coefficients for the socio-demographic factors (individual level and contextual level factors), perception, and attitudes were rs = 0.50, rp = 0.48, and ra = 0.45, respectively, and are acceptable.” with 

“We then determined the construct validity of the instrument using confirmatory factor analysis. Our confirmatory factor analysis results depicted good model fitness indices of: χ2 (34) = [53.800, p = 0.017]; root mean squared error approximation (RMSEA) = 0.038; comparative fit index (CFI) = 0.937; Tucker-Lewis index (TLI) = 0.916; and standardized root mean squared residual (SRMR) = 0.039.”

 Kindly prepare the tables in three-line format.

Response: Thank you for this comment. We have now presented all tables in three-line format in line with APA 7 guidelines.

 Line 321: Table 5 might not need to be included in the main manuscript. If the authors would like to report, it would be better to include it as an appendix table.

Response: Thank you for the comment. Table 5 presents the composite distribution statistics of the healthcare professionals’ perception and attitudes towards COVID-19 vaccination. These statistics are central to the interpretation of overall perception and attitudes of the healthcare professionals. Therefore, we think that, it is necessary to keep Table 5 within the main text of the manuscript for easy access and reference.

---

## [Editor Report · Decision Letter 1]

31 Jan 2024

Healthcare professionals' perception and COVID-19 vaccination attitudes in North-Western Ghana: a multi-center analysis

PONE-D-23-35761R1

Dear Dr. Balegha

We’re pleased to inform you that your manuscript has been judged scientifically suitable for publication and will be formally accepted for publication once it meets all outstanding technical requirements.

Kind regards,

Pracheth Raghuveer, MD, DNB

Academic Editor

PLOS ONE
---

## [Editor Report · Acceptance letter]

12 Feb 2024

PONE-D-23-35761R1 

PLOS ONE

Dear Dr. Balegha, 

I'm pleased to inform you that your manuscript has been deemed suitable for publication in PLOS ONE. Congratulations! Your manuscript is now being handed over to our production team.

Kind regards, 

on behalf of

Dr. Pracheth Raghuveer 

Academic Editor

PLOS ONE